# Regulation of mTORC1 by lysosomal calcium and calmodulin

Ruo-Jing Li[1,2], Jing Xu[1,2,3], Chenglai Fu[4], Jing Zhang[5], Yujun George Zheng[5], Hao Jia[6], Jun O Liu[1,2,7]*

[1]Department of Pharmacology and Molecular Sciences, Johns Hopkins University School of Medicine, Baltimore, United States; [2]The SJ Yan and HJ Mao Laboratory of Chemical Biology, Johns Hopkins University School of Medicine, Baltimore, United States; [3]Eli Lilly and Company, Indianapolis, United States; [4]The Solomon H Snyder Department of Neuroscience, Johns Hopkins University School of Medicine, Baltimore, United States; [5]Department of Pharmaceutical and Biomedical Sciences, College of Pharmacy, The University of Georgia, Athens, United States; [6]Department of Physiology, Johns Hopkins University School of Medicine, Baltimore, United States; [7]Department of Oncology, Johns Hopkins University School of Medicine, Baltimore, United States

**Abstract** Blockade of lysosomal calcium release due to lysosomal lipid accumulation has been shown to inhibit mTORC1 signaling. However, the mechanism by which lysosomal calcium regulates mTORC1 has remained undefined. Herein we report that proper lysosomal calcium release through the calcium channel TRPML1 is required for mTORC1 activation. TRPML1 depletion inhibits mTORC1 activity, while overexpression or pharmacologic activation of TRPML1 has the opposite effect. Lysosomal calcium activates mTORC1 by inducing association of calmodulin (CaM) with mTOR. Blocking the interaction between mTOR and CaM by antagonists of CaM significantly inhibits mTORC1 activity. Moreover, CaM is capable of stimulating the kinase activity of mTORC1 in a calcium-dependent manner *in vitro*. These results reveal that mTOR is a new type of CaM-dependent kinase, and TRPML1, lysosomal calcium and CaM play essential regulatory roles in the mTORC1 signaling pathway.

*For correspondence: joliu@jhu.edu

**Competing interests:** The authors declare that no competing interests exist.

## Introduction

Mechanistic target of rapamycin (mTOR) plays an essential role in sensing a myriad of environmental cues including nutrients and growth factor stimulation to regulate cell growth and proliferation (*Wullschleger et al., 2006*). mTOR independently associates with raptor or rictor to form two distinct complexes, mTORC1 and mTORC2, respectively. The two complexes share several common subunits, including the catalytic mTOR subunit, mLST8, DEPTOR, and the Tti1/Tel2 complex (*Laplante and Sabatini, 2012*). Among the remaining components, PRAS40 are specific to mTORC1, whereas rictor, mSin1 protor1/2 are unique to mTORC2 (*Laplante and Sabatini, 2012*). These two complexes differ in their sensitivity to rapamycin, upstream signals and downstream outputs (*Laplante and Sabatini, 2012*). The mTORC1 complex integrates different extracellular and intracellular signal inputs, such as growth factors, amino acids, stress and energy status, to regulate cellular processes such as protein and lipid synthesis and autophagy, by phosphorylating and activating p70 S6 kinase (p70S6K) (*Chung et al., 1992*; *Price et al., 1992*) and eukaryotic translation initiation factor 4E-binding protein 1 (4E-BP1) (*Lin et al., 1995*; *von Manteuffel et al., 1996*). In contrast, mTORC2 is involved in Akt phosphorylation and regulation of the cellular cytoskeleton (*Bhaskar and*

*Hay, 2007*). Activation of mTORC1 by amino acids requires the translocation of mTORC1 from the cytosol to the surface of lysosomes, which is dependent on the Rag GTPase heterodimers RagA/B and RagC/D (*Kim et al., 2008*; *Sancak et al., 2008*).

The second messenger calcium has been shown to play an important role in the regulation of mTOR signaling. Earlier hints that calcium might be involved in mTOR signaling came from observations that calcium was required for the activation of p70S6K (*Conus et al., 1998*; *Graves et al., 1997*; *Hannan et al., 2003*). But the underlying mechanism was attributed to upstream regulators such as PI3K or isoforms of PKC. More definitive roles of calcium and its signaling mediator calmodulin (CaM) in mTORC1 signaling were demonstrated in the context of amino acid activation of the pathway (*Gulati et al., 2008*). It was shown that the phosphorylation of S6K1 in response to amino acids was inhibited by the cell permeable calcium chelator BAPTA-AM while thapsigargin, which releases intracellular calcium, activated mTORC1 activity. Moreover, it was shown that the activation of mTORC1 by amino acids was inhibited by antagonists of CaM or its knockdown using siRNA, suggesting that CaM is required for mTORC1 activity. The underlying mechanism by which calcium and CaM regulate mTORC1 was attributed to the binding of calcium-activated CaM to the hVps34, leading to the activation of its kinase activity. While the sensitivity of mTORC1 to BAPTA-AM and CaM antagonists have been reproducibly observed, ensuing studies have cast some doubt on the notion that hVps34 is a key mediator of calcium and CaM in the regulation of mTORC1 in similar and other cellular systems (*Mercan et al., 2013*; *Yan et al., 2009*).

In a previous study, we found that small molecules that are known to induce Niemann-Pick Disease Type C (NPC) phenotype inhibited mTOR (*Xu et al., 2010*). Independently, it has also been reported that NPC cells showed significant defects in lysosomal calcium homeostasis (*Lloyd-Evans et al., 2008*; *Shen et al., 2012*). Cells that have mutations in or deficient mucolipin transient receptor potential (TRP) channel 1 (TRPML1) display altered $Ca^{2+}$ homoeostasis similar to that seen in NPC cells (*Cheng et al., 2010*; *Dong et al., 2010*; *Shen et al., 2011*). Cells treated with chemical NPC inducers exhibited reduced TRPML1-mediated lysosomal $Ca^{2+}$ release in response to a TRPML1 agonist, indicating dysfunction of this calcium channel. Furthermore, it has been shown that TRPML1 homolog in fly is required for TORC1 activation and fusion of amphisomes with lysosomes, and the inhibition of TORC1 can be rescued by feeding fly larvae with a high-protein diet (*Wong et al., 2012*; *Venkatachalam et al., 2013*). Furthermore, TORC1 also exerts reciprocal control on TRPML function, establishing the connection between TRPML and TORC1 signaling pathway in fly cells. Whether TRPML1 regulates mTORC1 signaling pathway in mammalian cells remains unknown. Putting these findings together, we hypothesized that a defect in lysosomal calcium homeostasis in NPC cells might be responsible for the observed inhibition of the mTOR signaling pathway.

We validated our hypothesis by demonstrating that depletion of TRPML1 inhibits mTORC1 while overexpression or pharmacologic activation of TRPML1 activates mTORC1. We traced the likely site of regulation of mTORC1 pathway by calcium and CaM by determining the sensitivity of mTORC1 activity to BAPTA-AM and CaM antagonists in response to various upstream activators of the kinase complex and narrowed it to mTORC1 itself. We found that CaM interacted with mTORC1 and activated its kinase activity. Together, these findings shed significant new light on mTORC1 signaling pathway and offer a unifying mechanism that accounts for most, if not all, earlier observations implicating calcium and CaM in the regulation of mTORC1 by different upstream activators in distinct cellular context.

## Results

### TRPML1 is required for the activation of mTORC1

To determine whether TRPML1 is required for mTORC1 signaling, HEK293T cells were transduced with lentiviral shRNA targeting human TRPML1 (Sh1 and Sh2) or a scrambled shRNA (Scr). Due to lack of reliable hTRPML1 antibodies, the knockdown efficiency was assessed by RT-qPCR (*Figure 1a*, bottom panel) as well as indirectly by the expression level of ectopically expressed EGFP-TRPML1. The activity of mTORC1, as judged by the phosphorylation of S6K, was significantly inhibited upon TRPML1 knockdown, while the phosphorylation of Akt (T308) was not affected (*Figure 1a*). To determine whether the mTOR inhibition caused by TRPML knockdown was due to blockade of lysosomal calcium release, we performed a rescue experiment in Human Umbilical Vein Endothelial Cells

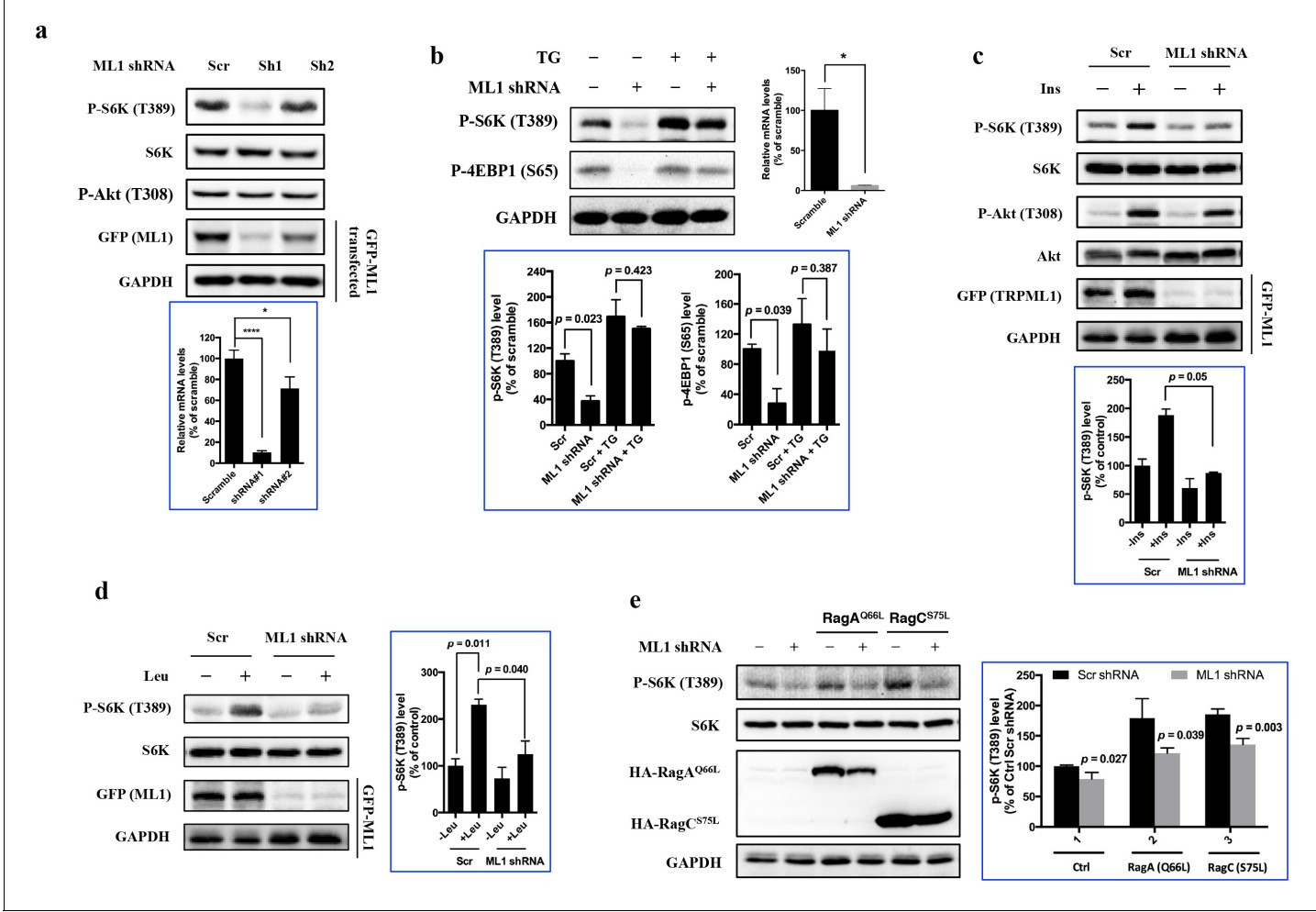

**Figure 1.** TRPML1 is required for the full activation of mTORC1. (**a**) HEK293T cells were transduced with lentiviral scrambled shRNA (Scr) and shRNA targeting human TRPML1 (Sh1 and Sh2), respectively. To assess the knockdown efficiency, a fraction of transduced cells were transfected with EGFP-TRPML1. After 24 hr, transfected or untransfected cells were lysed and subjected to immunoblotting. Untransfected cells were used to detect p-S6K, S6K and p-Akt, and transfected cells were used to detect GFP and GAPDH. RT-qPCR was also performed to evaluate the knockdown efficiency (bottom panel) (mean ± s.d., n = 2 independent experiments). (**b**) Scrambled shRNA or TRPML1 shRNA-transduced HUVEC were treated with vehicle control or thapsigargin (5 μM) for an additional 2 hr. Cells were lysed and subjected to immunoblotting. Knockdown efficiency was assessed by RT-qPCR (right panel). The bottom panel of plots shows the percentage of p-S6K and p-4EBP1 levels compared with scramble shRNA transduced vehicle control treated HUVEC normalized by GAPDH loading control (mean ± s.d., n = 2 independent experiments). (**c** and **d**) Scrambled shRNA or TRPML1 shRNA transduced HEK293T cells were deprived for 24 hr of serum (**c**) or 3 hr of leucine (**d**) and, where indicated, were stimulated with 600 nM insulin or 52 μg/ml leucine for 10 min. Simultaneously, another fraction of scrambled shRNA or TRPML1 shRNA-transduced cells were transfected with EGFP-TRPML1 for 24 hr. Cells were lysed and subjected to immunoblotting. The plots show the percentage of p-S6K levels compared with scramble shRNA transduced serum (**c**) or leucine (**d**) starved HEK293T cells normalized by total S6K control (mean ± s.d., n = 2 independent experiments, respectively). (**e**) Scrambled shRNA or TRPML1 shRNA-transduced HEK293T cells were transfected with Rag AQ66A or Rag CS75L for 24 hr. Cells were lysed and subjected to immunoblotting. The plot shows the percentage of p-S6K level compared with scramble shRNA transduced empty vector transfected HEK293T cells normalized by total S6K control. (mean ± s.d. for n = 3 independent experiments).

The following figure supplement is available for figure 1:

**Figure supplement 1.** Effects of TRPML1 on mTORC1 activation.

(HUVEC) (*Figure 1b*) and HEK293T (*Figure 1—figure supplement 1a*) using thapsigargin, a sarco/endoplasmic reticulum $Ca^{2+}$-ATPase inhibitor that increases cytosolic calcium concentrations (*Lytton et al., 1991*). Indeed, the inhibition of mTORC1 activity by TRPML1 knockdown was rescued by thapsigargin, suggesting that mTORC1 inhibition was due, in large part, to the lack of lysosomal

calcium release. Moreover, knocking down TRPML1 also attenuated the activation of mTORC1 by insulin (*Figure 1c*), leucine (*Figure 1d*) as well as overexpression of constitutively active RagA or RagC (*Figure 1e*). Furthermore, we determined the phosphorylation of S6K in normal human fibroblasts (TRPML1 +/+) and fibroblasts from a mucolipidosis IV patient (TRPML1 -/-). Compared with TRPML1 +/+ human fibroblasts, TRPML1 -/- cells showed decreased phosphorylation of S6K (*Figure 1—figure supplement 1b*). Interestingly, this inhibition was partially reversed by leucine compared with that in wild type cells (*Figure 1—figure supplement 1c*). However, the treatment of thapsigargin fully restored the phosphorylation of S6K (*Figure 1—figure supplement 1b*), suggesting that in mammalian cells, the decrease in mTORC1 activity in TRPML1 mutant cells is not only due to the incomplete autophagy that has been reported in *Drosophila* (*Wong et al., 2012*). In addition, knockdown of other lysosomal channels, such as TPC2 and P2X4, did not significantly inhibit mTORC1 signaling (*Figure 1—figure supplement 1d*), indicating that the decreased mTOR activity upon TRPML1 knockdown was not due to the dysregulation of the structure of the endolysosmal system, and as one of the lysosomal calcium channels, TRPML1 may play a more dominant role in the regulation of mTORC1 signaling.

Having shown that TRPML1-mediated lysosomal calcium release is necessary for mTORC1 activity, we then turned to the reciprocal question of whether an increase in lysosomal calcium release through TRPML1 could stimulate mTORC1. Thus, HEK293T cells were transfected with expression plasmids for EGFP-TRPML1 and its non-conducting pore mutant (D471K/D472K) EGFP-TRPML1 (KK), respectively. The phosphorylation of S6K was slightly but significantly increased by overexpression of wild type TRPML1 but not the non-conducting pore mutant TRPML1 (KK) (*Figure 2a*). Next, we treated HEK293T cells with TRPML1 agonist MLSA1 (*Shen et al., 2012*; *Feng et al., 2014*). The phosphorylation of S6K was increased by MLSA1 in a dose-dependent manner (*Figure 2b*). In contrast, MLSA1 failed to increase the phosphorylation of S6K in the cells transduced with lentiviral TRPML1 shRNA, or pretreated with bafilomycin A1 or Glycyl-L-phenylalanine 2-naphthylamide (GPN), suggesting that the increase in S6K phosphorylation induced by MLSA1 was mediated by calcium released through TRPML1 (*Figure 2c,d*). Upon amino acids stimulation, mTOR translocated from the cytosol to the lysosome, colocalizing with EGFP-TRPML1 (*Figure 2—figure supplement 1*), indicating that the activation of TRPML1 acted independently of the translocation of mTORC1 induced by amino acids. The upregulated TRPML1 has been reported to promote autophagy (*Wang et al., 2015*; *Medina et al., 2015*; *Wong et al., 2012*). To determine if activation of mTORC1 in response to TRPML1 overexpression was due to up-regulated autophagy, we overexpressed constitutively active Rab7A (Q67L) and dominant negative Rab7A (T22N) in HEK293T cells (*Jager et al., 2004*; *Hyttinen et al., 2013*). As shown in *Figure 2—figure supplement 2*, overexpression of neither constitutively active nor dominant negative Rab7A affected mTORC1 signaling, while overexpression of TRPML1 plus MLSA1 treatment stimulated the phosphorylation of S6K, suggesting that the activated mTORC1 by TRPML1 stimulation was not mediated through autophagy.

## Calcium and CaM are required for activation of mTORC1

Both intracellular calcium and CaM have been reported to be required for mTORC1 activity (*Conus et al., 1998*; *Graves et al., 1997*; *Gulati et al., 2008*; *Hannan et al., 2003*; *Mercan et al., 2013*). We thus treated HEK293T cells with the cytosolic $Ca^{2+}$ chelator BAPTA-AM (BAPTA) or the CaM antagonists W-7 and calmidazolium (CMDZ). In agreement with previous studies (*Gulati et al., 2008*; *Ke et al., 2013*; *Graves et al., 1997*; *Zhou et al., 2010*), we observed that BAPTA , W-7 and CMDZ inhibited phosphorylation of S6K in a dose-dependent manner with $IC_{50}$ values of 3.96 ± 1.30 μM, 21.59 ± 1.81 μM and 10.36 ± 0.59 μM, respectively (*Figure 3—figure supplement 1*). In comparison to S6K, phosphorylation of Akt (S473), the substrate of mTORC2, was also inhibited by CMDZ and W-7, but at much higher concentrations ($EC_{50}$ values of 27.21 ± 9.82 μM and 45.91 ± 9.61 μM, respectively) compared with that of p-S6K, while BAPTA did not show appreciable inhibition to p-Akt (S473) (*Figure 3—figure supplement 1c*). In addition, CMDZ and BAPTA also showed potent inhibitory effect on mTORC1 signaling pathway in HUVEC and A549 cells (*Figure 3—figure supplement 2a,b*), suggesting that mTORC1 is also regulated by $Ca^{2+}$/CaM in primary and other cancer cells.

Next, we investigated how fast mTORC1 and mTORC2 responded to BAPTA or CMDZ. As shown in *Figure 3a*, both CMDZ and BAPTA caused appreciable inhibition of mTORC1 activity within 0.5–1 hr as judged by the phosphorylation of S6K and 4EBP1. In contrast, the phosphorylation of Akt

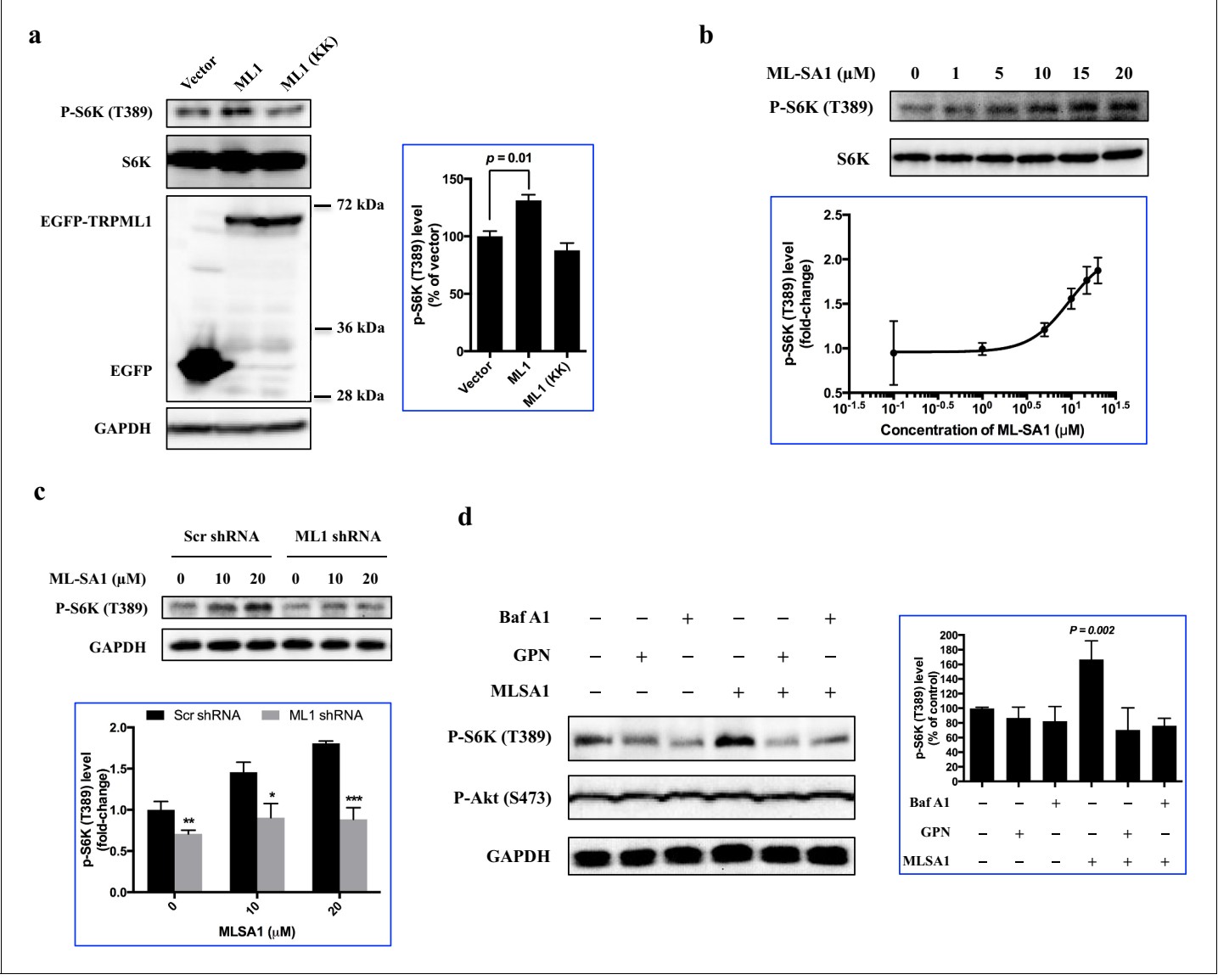

**Figure 2.** Overexpression or pharmacological stimulation of TRPML1 activates mTORC1 signaling pathway. (a) HEK293T cells (80% confluency) were transfected with EGFP vector, EGFP-TRPML1 or its non-conducting pore mutant (D471K/D472K) EGFP-TRPML1 (KK) for 20 hr. Cells were lysed and subjected to immunoblotting. The plot shows the percentage of p-S6K levels compared with vector transfected cells normalized by total S6K control (mean ± s.d., n = 3 independent experiments). (b) HEK293T cells were treated with different concentrations of ML-SA1 for 3 hr. Cells were lysed and subjected to immunoblotting. The plot shows the dose-response curve of ML-SA1 normalized by total S6K. (c) Scrambled shRNA or TRPML1 shRNA transduced HEK293T cells were treated with varying concentrations of MLSA1 for 3 hr. Cells were lysed and subjected to immunoblotting. The plot shows the percentage of p-S6K levels compared with scramble shRNA transduced vehicle control treated 293T cells normalized by GAPDH loading control (mean ± s.d., n = 3 independent experiments). (d) HEK293T cells were pretreated with bafilomycin A1 (1 μM) or GPN (200 μM) for 1 hr, followed by treatment with or without MLSA1 for an additional 1.5 hr. Cells were lysed and subjected to immunoblotting. The plot shows the percentage of p-S6K levels compared with vehicle control normalized by GAPDH loading control (mean ± s.d., n = 3 independent experiments). *p<0.05, **p<0.01, n.s. no significant difference.

The following figure supplements are available for figure 2:

**Figure supplement 1.** Colocalization of EGFP-TRPML1 and mTOR.

**Figure supplement 2.** Effects of constitutively active or dominant negative Rab 7A.

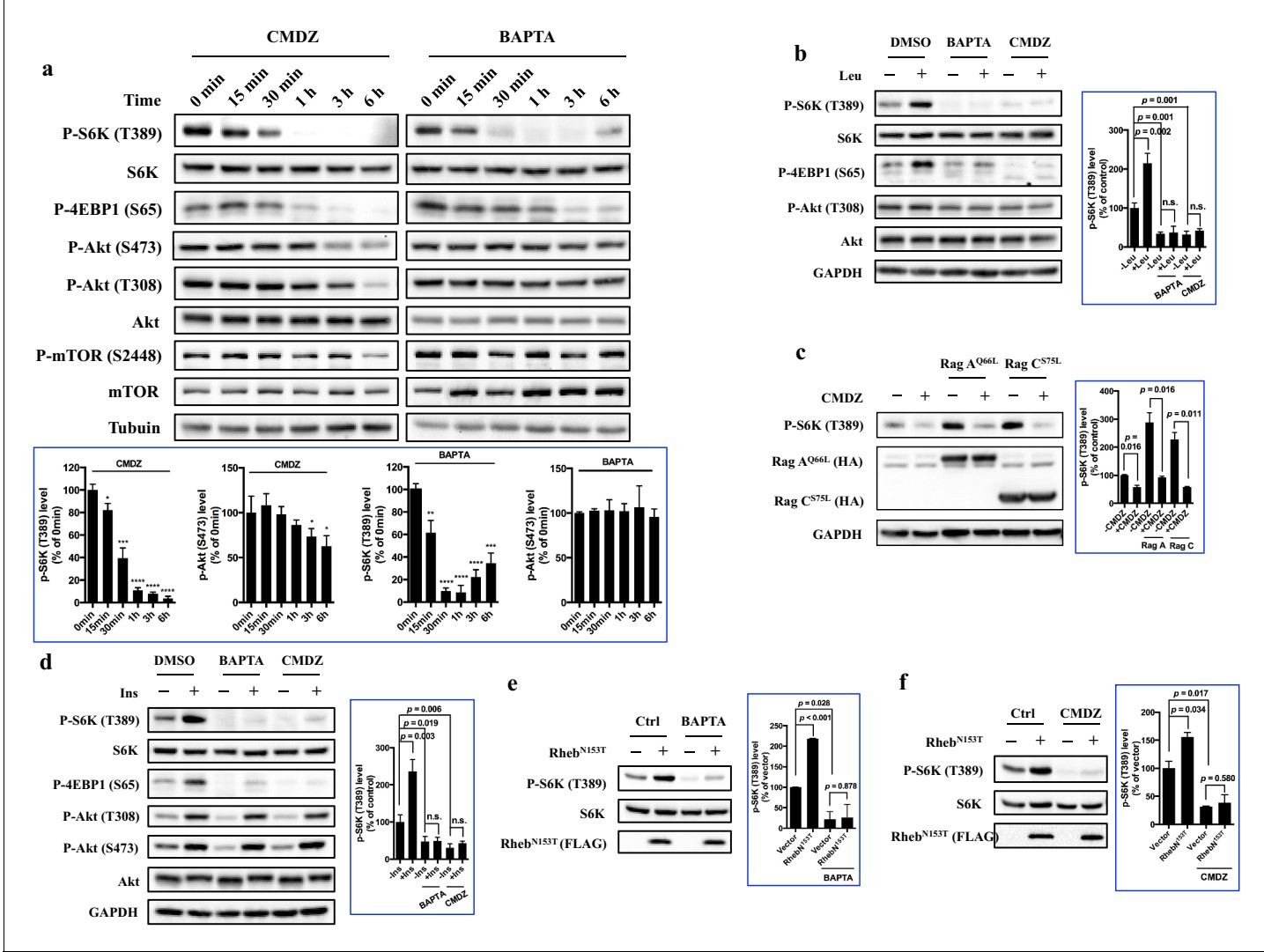

**Figure 3.** Regulation of mTORC1 by cytosolic calcium and CaM occurs proximal to mTORC1 itself. (a) Effects of 10 µM CaM antagonist calmidazolium (CMDZ) or 25 µM cytosolic $Ca^{2+}$ chelator BAPTA-AM (BAPTA) on the phosphorylation of different proteins of the mTOR signaling pathway at different time points. The bottom panel of plots shows the percentage of p-S6K and p-Akt (S473) levels compared with 0 min treated 293T cells normalized by total S6K and total Akt control, respectively (mean ± s.d., n = 3 independent experiments). (b and d) Effects of CMDZ (10 µM) or BAPTA-AM (25 µM) on the phosphorylation of indicated proteins in response to deprivation and stimulation with leucine (b) and insulin (d). Cell lysates were prepared from HEK293T cells deprived for 3 hr of leucine (b) or 24 hr of serum (d) and, where indicated, stimulated with 52 µg/ml leucine or 600 nM insulin for 10 min. CMDZ (10 µM) or BAPTA-AM (25 µM) was added 1 hr prior to cell harvesting. The plots show the percentage of p-S6K levels compared with vehicle control treated leucine (b) or serum (d) starved HEK293T cells normalized by total S6K control (mean ± s.d., n = 3 independent experiments, respectively). (c) Effects of CMDZ (10 µM) on the phosphorylation of S6K in HEK293T cells transfected with constitutively active RagA or RagC in expression vectors. Cell lysates were prepared and subjected to immunoblotting. The plot shows the percentage of p-S6K levels compared with vector transfected vehicle control treated 293T cells normalized by GAPDH loading control (mean ± s.d., n = 3 independent experiments). (e and f) Effects of 25 µM BAPTA-AM (e) and 10 µM CMDZ (f) on the phosphorylation state of S6K in HEK293T cells stably expressing constitutively active Rheb as indicated. HEK293T cells were transduced with lentiviral FLAG-tagged Rheb$^{N153T}$, and treated with indicated compounds for 1 hr. Cell lysates were prepared and used for immunoblotting. The plots show the percentage of p-S6K levels compared with vehicle control treated empty lentiviral vector transduced 293T cells normalized by total S6K control (mean ± s.d., n = 2 independent experiments). *p<0.05, **p<0.01, ***p<0.001, ****p<0.0001, n.s. no significant difference.

The following figure supplements are available for figure 3:

**Figure supplement 1.** Effects of calmodulin antagonist (a) calmidazolium (CMDZ), (b) W-7 and (c) cytosolic $Ca^{2+}$ chelator BAPTA-AM on phosphorylation state of S6K (T389) and Akt (S473).

*Figure 3 continued on next page*

*Figure 3 continued*

**Figure supplement 2.** Effects of calmodulin antagonist calmidazolium (CMDZ) and cytosolic $Ca^{2+}$ chelator BAPTA-AM on mTOR signaling pathway in HUVEC and A549 cells.

(T308) and its substrate, mTOR (S2448), was not significantly affected by CMDZ until 6 hr post treatment. In addition, CMDZ did not cause significant inhibition of phosphorylation of Akt (S473) until 3 hr after treatment, indicating that the response of mTORC2 to the CaM antagonist has a much slower onset than that of mTORC1 (*Figure 3a*, left panel). Although the onset of the effect of BAPTA on 4EBP1 phosphorylation was slightly slower than that of CMDZ, BAPTA did not significantly affect the phosphorylation of either Akt (S473, T308) or mTOR even after 6 hr (*Figure 3a*, right panel). These results suggest that CaM regulates both mTORC1 and mTORC2, but the two complexes differ in their sensitivity to CaM and calcium. Interestingly, after a 6-hr treatment with BAPTA, the inhibition of phosphorylation of S6K and 4EBP1 was partially reversed, which is consistent with a previous report that over time upon treatment with BAPTA, a gradual increase in intracellular $Ca^{2+}$ was seen (*Wei et al., 1998*). The relatively short time required for CMDZ and BAPTA to exert their effects on mTORC1 and their faster onset than those on mTORC2 suggested that the inhibition of mTORC1 likely occurred independently of its upstream signaling events, such as phosphorylation of Akt (T308).

## Regulation of mTORC1 by cytosolic calcium and CaM occurs proximal to mTORC1 itself

To further explore the level at which $Ca^{2+}$ and CaM regulate mTORC1 signaling, we determined the effects of CMDZ and BAPTA on mTORC1 activation in response to various upstream activating stimuli of mTORC1. Similar to previous observations (*Gulati et al., 2008*), we found that leucine-stimulated mTORC1 activation was inhibited by BAPTA and CMDZ (*Figure 3b*, Lanes 4 vs. 2 and 6 vs. 2). The activation of mTORC1 by leucine has been shown to be mediated by the small GTPases RagA/B and RagC/D (*Kim et al., 2008*), and overexpression of constitutively active $RagA^{Q66L}/RagC^{S75N}$ can bypass leucine to activate mTORC1. We found that activation of mTORC1 by either $RagA^{Q66L}$ or $RagC^{S75N}$ remained sensitive to CMDZ (*Figure 3c*). Next, we determined whether activation of mTORC1 by insulin was also sensitive to CaM blockade. Although insulin strongly increased the phosphorylation of Akt (T308) (*Figure 3d*, Lanes 2), the mTORC1 activity remained sensitive to CMDZ as well as BAPTA-AM (*Figure 3d*, Lanes 4 and 6). It has been reported that mTOR is directly bound to and activated by Rheb-GTP (*Long et al., 2005*). Thus, we used HEK293T, HUVEC and A549 to produce stable cell lines overexpressing constitutively active $Rheb^{N153T}$ as previously described (*Yan et al., 2006*), and determined their sensitivity to BAPTA and CMDZ. $Rheb^{N153T}$-induced phosphorylation of S6K remained sensitive to inhibition by BAPTA and CMDZ in HEK293T, HUVEC and A549 cells (*Figure 3e,f*, *Figure 3—figure supplement 2c,d*). Together, these results suggested that the site of regulation of mTORC1 by $Ca^{2+}$ and CaM lies proximal to mTORC1 itself.

## CaM interacts with mTOR

CaM has been previously reported to indirectly interact with mTORC1, and human vacuolar protein sorting 34 (hVps34) was shown to mediate the interaction between CaM and mTORC1 in HeLa cells (*Gulati et al., 2008*). To our surprise, when hVps34 was knocked down in HEK293T cells, binding of CaM to mTOR was not affected (*Figure 4a*), neither was the sensitivity of mTORC1 to CaM (*Figure 4—figure supplement 1*), ruling out hVps34 as a mediator of CaM-mTOR interaction in HEK293T cells. These results raised the possibility that CaM may directly interact with a subunit of the mTORC1 complex, thereby regulating its kinase activity. Indeed, CaM sepharose could pull down mTOR and raptor, but not PRAS40, in a $Ca^{2+}$-dependent manner (*Figure 4b*). The interaction between mTOR and CaM was sensitive to detergents and the CaM antagonist W-7 (*Figure 4b* and *Figure 4—figure supplement 2*). However, $Ca^{2+}$ did not affect the assembly of mTORC1 complex (*Figure 4—figure supplement 3*), suggesting that one of the interactions of CaM sepharose with mTOR and raptor could be indirect. To identify the subunit in mTORC1 that interacts with CaM, we knocked down raptor and mTOR, respectively, and determined the remaining interaction between

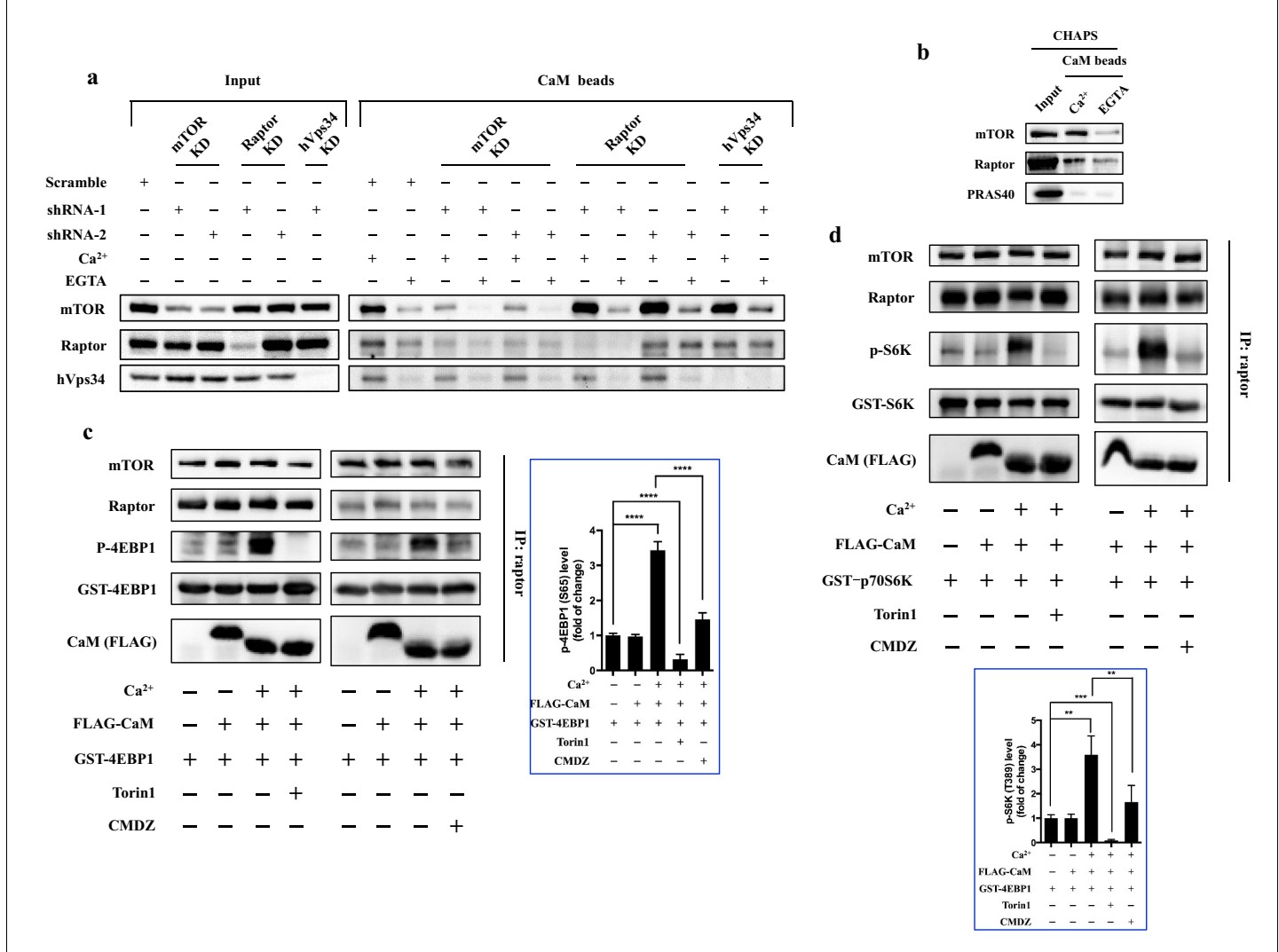

**Figure 4.** CaM interacts with mTORC1 and regulates mTORC1 kinase activity in vitro. (**a**) CaM interacts with mTOR independent of hVps34 or raptor. Cell lysates were prepared from HEK293T cells transduced with lentiviral shRNAs targeting human mTOR, raptor, hVps34 or scrambled shRNA, followed by CaM sepharose precipitation in the presence of $CaCl_2$ (1 mM) or EGTA (5 mM). The cell lysates and precipitates were analyzed by immunoblotting to detect the indicated proteins. (**b**) Endogenous mTORC1 was pulled down by CaM sepharose in a $Ca^{2+}$-dependent manner. HEK293T cells were lysed in CHAPS buffer, and the lysates were incubated with CaM sepharose in the presence of $CaCl_2$ (1 mM) or EGTA (5 mM). The precipitates were analyzed by immunoblotting. (**c** and **d**) Cell lysates were prepared from HEK293T cells in CHAPS buffer, and endogenous mTORC1 was immunoprecipitated by a raptor antibody. ATP (250 μM), Torin1 (100 nM), CMDZ (8 μM), CaM (2 μM) or/and $CaCl_2$ (1 mM) were added into the kinase reaction as indicated. Phosphorylation of 4EBP1 (**c**) and S6K (**d**) were detected by immunoblotting. The plots show the fold of change of phosphorylation of 4EBP1 (**c**) or S6K (**d**) compared with control group (first lane) normalized by total GST-tagged protein control. (mean ± s.d., n = 6 and 5 independent experiments, respectively).

The following figure supplements are available for figure 4:

**Figure supplement 1.** Effects of calmidazolium (CMDZ) on hVps34 depleted cells.

**Figure supplement 2.** CaM interacts with mTORC1 independently of raptor or hVps34.

**Figure supplement 3.** The presence or absence of $Ca^{2+}$ does not affect the association of mTORC1.

**Figure supplement 4.** Proposed model of regulation of mTORC1 by TRPML1, lysosomal calcium and CaM.

each subunit and CaM sepharose (*Figure 4a*). Knockdown of raptor had no effect on the pulldown of mTOR by CaM sepharose. In contrast, knockdown of mTOR significantly reduced the binding of raptor as well as mTOR to CaM (*Figure 4a*), suggesting that the interaction between mTOR and CaM is independent of raptor.

## Ca$^{2+}$ and CaM activate the kinase activity of isolated mTORC1 complex *in vitro*

Having shown that CaM binds to mTORC1, we asked the question of whether CaM and Ca$^{2+}$ had a direct effect on the intrinsic kinase activity of isolated mTORC1 complex *in vitro*. Thus, endogenous mTORC1 complex was immunoprecipitated by an anti-raptor antibody, and an *in vitro* kinase assay was performed using purified recombinant 4EBP1 as a substrate (*Sarbassov et al., 2004*). As shown in *Figure 4c*, the phosphorylation of 4EBP1 by immunoprecipitated mTORC1 complex was significantly increased in the presence of both CaCl$_2$ (1 mM) and CaM (2 μM), but not CaM alone, indicating that CaM activates mTORC1 kinase activity *in vitro* in a Ca$^{2+}$-dependent manner (*Figure 4c*, top left and top right panels, Lanes 1–3, respectively). Importantly, the activation of mTORC1 by Ca$^{2+}$/CaM was inhibited by Torin 1 (*Figure 4c*, top left panel, Lane 4), a TOR kinase inhibitor, and CMDZ (*Figure 4c*, top right panel, Lane 4), indicating that the phosphorylation of 4EBP1 was dependent on TOR kinase activity and CaM. Similar results were obtained from the *in vitro* kinase assay using purified recombinant S6K as the substrate (*Figure 4d*). Together, these results demonstrated that binding of CaM to mTORC1 leads to the stimulation of kinase activity of the mTORC1 complex.

## Discussion

The work described in this manuscript reveals a novel mechanism of regulation of mTORC1 by lysosomal calcium and CaM (*Figure 4—figure supplement 4*), shedding new light on the mTOR signaling pathway. In the current model of mTORC1 activation (*Dibble and Cantley, 2015*; *Buerger et al., 2006*; *Efeyan and Sabatini, 2013*; *Saito et al., 2005*), growth factors, energy, and other inputs signal to mTORC1 primarily through the TSC-Rheb axis; amino acids act by regulating the nucleotide state of the heterodimeric Rag GTPases and promoting the translocation of mTORC1 onto lysosomes, where it interacts with and becomes activated by lysosomally-localized, GTP-bound Rheb (*Sancak et al., 2008*). Our results have uncovered another role of lysosomal localization of mTORC1, i.e., to receive localized lysosomal calcium stimulation. Integrating our previous observations (*Xu et al., 2010*) and the results from the present study, we propose an addition to the current model of mTOR signaling pathway: upon the translocation of mTORC1 onto the lysosome, properly released lysosomal calcium enriches local Ca$^{2+}$ concentration, prompting Ca$^{2+}$ binding to a local population of CaM, which in turn binds mTORC1 and stimulates the kinase activity of the mTORC1 complex.

The depletion of the homolog of TRPML1, TRPML in *Drosophila*, results in decreased TORC1 signaling, which was attributed to incomplete autophagy, and was completely reversed by feeding fly larvae with a high-protein diet (*Wong et al., 2012*; *Venkatachalam et al., 2013*). However, we showed that in TRPML1-knockdown mammalian cells or mucolipidosis IV human fibroblasts, the inhibited mTORC1 signaling was only partially reversed by leucine or overexpression of constitutively active Rag GTPase, suggesting that there is a difference in the mechanisms of regulation of mTOR by Ca$^{2+}$/CaM between mammalian and fly cells. Interestingly, thapsigargin, which increases cytosolic Ca$^{2+}$, could completely restore phosphorylation of S6K in TRPML1 deficient cells to the control level. Given that increased cytosolic Ca$^{2+}$ also positively regulates the Ca$^{2+}$-dependent fusion of late-endosomes and autophagosomes to lysosomes (*Grotemeier et al., 2010*; *Lloyd-Evans et al., 2008*; *Wong et al., 2012*), the rescue effect of thapsigargin might be due to the combined effects of autophagy as well as the direct stimulation of mTORC1 by Ca$^{2+}$/CaM in mammalian cells. On the other hand, TRPML1 is significantly upregulated under amino acids starvation (*Wang et al., 2015*), when mTORC1 dissociates from the lysosomal surface and becomes inactive, indicating that mTORC1 and TRPML1 may form reciprocal regulation loop. In addition, it has been recently reported that under starvation, lysosomal Ca$^{2+}$ release through TRPML1 activates local calcineurin, a Ca$^{2+}$, CaM-dependent protein phosphatase, which dephosphorylates TFEB and promotes its nuclear translocation as well as regulates lysosomal biogenesis (*Medina et al., 2015*), suggesting another local function of lysosomal Ca$^{2+}$.

Our model of regulation of mTORC1 by $Ca^{2+}$ and CaM differs from that proposed in a previous report (*Gulati et al., 2008*), even though some of the experimental observations are in agreement. Similar to previous reports (*Gulati et al., 2008*; *Ke et al., 2013*; *Graves et al., 1997*), we found that mTORC1 activity is sensitive to inhibition by BAPTA-AM and CaM antagonist CMDZ (*Figure 3*), suggesting that both intracellular calcium and CaM are required for mTORC1 activation. However, the precise mechanism of regulation of mTORC1 by calcium and CaM is distinct in our new model. First, we demonstrated that the lysosomal pool of calcium plays a unique and critical role in mTORC1 activation in mammalian cells. In earlier studies, however, the sources of calcium have been only suggested to be extracellular (*Conus et al., 1998*; *Gulati et al., 2008*) or conventional intracellular calcium stores such as the ER (*Ke et al., 2013*; *Graves et al., 1997*; *Zhou et al., 2010*). Second, a previous study showed the CaM associates with mTORC1 complex through hVps34, and calcium and CaM activate mTORC1 via hVps34 activation (*Gulati et al., 2008*). In an independent study, it was shown that hVps15, but not $Ca^{2+}$/CaM, activates hVps34 (*Yan et al., 2009*). Similarly, we also found that knockdown of hVps34 had no effect on the interaction between CaM and mTORC1 in HEK293T cells, ruling out involvement of hVps34 in the regulation of mTORC1 via $Ca^{2+}$/CaM, at least in this cell type. We surmise that most of the previous results implicating calcium or CaM in the regulation of mTORC1 may be explained by our current model.

In previous studies, *in vitro* kinase assay of mTORC1 used EDTA in immunoprecipition buffer (*Kim et al., 2002*, *2003*), which precluded the detection of any regulatory effect of calcium and CaM. By performing the mTOR kinase assay in the absence or presence of calcium and CaM *in vitro*, we were able to observe a dramatic activation of mTORC1 by calcium and CaM, revealing the functional consequence of the binding of CaM to mTOR–activation of its intrinsic kinase activity. As such, mTOR is a new type of atypical CaM-dependent kinase. The newly uncovered roles of lysosomal calcium and CaM in the regulation of mTOR signaling not only fill a gap in our understanding of this fundamental signaling pathway, but also offer new molecular targets for discovering and developing novel mTOR inhibitors.

## Materials and methods

### Cell lines and tissue culture

HEK293T (RRID: CVCL_0063, purchased from ATCC, the identity has been authenticated using STR profiling) and A549 (RRID: CVCL_0023, purchased from ATCC, the identity has been authenticated using STR profiling) cells were cultured in low glucose DMEM (Life Technology) supplemented with 10% FBS (Life Technology). Healthy human fibroblasts (Coriell Insititute, GM03440) and mucolipidosis IV human fibroblasts (Coriell Insititute, GM02048) were cultured in EMEM (ATCC) supplemented with 15% FBS. HUVEC (purchased from Lonza) were cultured in EGM media (Lonza). All cells were cultured at 37°C with the presence of 5% $CO_2$. All cell lines were tested for mycoplasma contamination and showed negative result. HEK293T and A549 cells have been authenticated using STR profiling at Johns Hopkins Genetic Resources Core Facility. Match percent was searched and compared with American Tissue Culture Collection database. HEK293T cells showed 100% matching to ATCC HEK293T reference profiling (ATCC number CRL-3216), and A549 cells showed 93% matching to ATCC A549 reference profiling (ATCC number CCL-185). Given $\geq$ 80% level of matching indicates that the cell lines are related, we concluded that both HEK293T and A549 cell lines are authenticated.

### Leucine starvation and stimulation of the cells

Almost confluent cultures in 6-well plates were washed once with leucine-free low glucose DMEM (US Biological), incubated in leucine-free DMEM for 3 hr, and stimulated with 52 µg/ml leucine for 10 min. For those cells treated with calmidazolium (CMDZ, Cayman Chemical) or BAPTA-AM (Cayman Chemical), compounds were added 1 hr prior to cell harvesting. Cells were processed for biochemical assays as described below.

### Growth factor starvation and insulin stimulation of the cells

Almost confluent cultures in 6-well plates were washed once with FBS-free DMEM, incubated in FBS-free DMEM for 24 hr, and stimulated with 600 nM insulin (Life Technology) for 10 min. For those

cells treated with calmidazolium (CMDZ) or BAPTA-AM, compounds were added 1 hr prior to cell harvesting. Cells were processed for biochemical assays as described below.

## Immunoblotting analysis

After indicated treatments, cells were washed once with ice-cold PBS and lysed in ice-cold RIPA buffer (20 mM Tris-HCl (pH 7.5), 150 mM NaCl, 1 mM $Na_2EDTA$, 1 mM EGTA, 1% NP-40, 1% sodium deoxycholate, 2.5 mM sodium pyrophosphate, 1 mM beta-glycerophosphate, 1 mM $Na_3VO_4$, 1 µg/ml leupeptin). After brief sonication, the cell debris was removed by centrifugation at 13,000 rpm for 10 min in a microfuge, and the protein amount in the supernatant was quantified and mixed with a proper volume of 5x SDS loading buffer. Proteins were then separated by SDS-PAGE and transferred to nitrocellulose membranes. After blocking at room temperature for 1 hr, membranes were immunoblotted with anti-p-S6K (T389) (1:1000, Cell Signaling Technology, Cat. 9205, RRID: AB_330944), p-Akt (T308) (1:1000, Cell Signaling Technology, Cat. 9275, RRID: AB_329828), p-Akt (S473) (1:1000, Cell Signaling Technology, Cat. 9271, RRID: AB_329825), p-4EBP1 (S65) (1:1000, Cell Signaling Technology, Cat. 9451, RRID: AB_330947), p-mTOR (S2448) (1:1000, Cell Signaling Technology, Cat. 2971, RRID: AB_330970), mTOR (1:1000, Cell Signaling Technology, Cat. 2983, RRID: AB_2105622), Akt (1:1000, Cell Signaling Technology, Cat. 9272, RRID: AB_329827), raptor (1:1000, Cell Signaling Technology, Cat. 2280, RRID: AB_10830734), PRAS40 (1:1000, Cell Signaling Technology, Cat. 2610, RRID: AB_916206), hVps34 (1:1000, Cell Signaling Technology, Cat. 3811, RRID: AB_2062856), HA tag (1:1000, Santa Cruz Biotechnology, Cat. sc-7392, RRID: AB_627809), myc tag (1:1000, Santa Cruz Biotechnology, Cat. sc-40, RRID: AB_627268), FLAG tag (1:5000, Sigma, Cat. F1804, RRID: AB_262044), GFP (1:1000, Santa Cruz Biotechnology, Cat. sc-9996, RRID: AB_627695), S6K (1:1000, Santa Cruz Biotechnology, Cat. sc-8418, RRID: AB_628094), GAPDH (1:1000, Santa Cruz Biotechnology, Cat. sc-20357, RRID: AB_641107) at 4°C overnight with the primary antibodies, followed by incubation with HRP-conjugated anti-mouse (1:10000, GE Healthcare, Cat. NXA931, RRID: AB_772209), anti-rabbit (1:10000, GE Healthcare, Cat. NA934, RRID: AB_772206) or anti-goat IgG (1:10000, Santa Cruz Biotechnology, Cat. sc-2020, RRID: AB_631728) at room temperature for 1 hr. Antibody-protein complexes were detected using enhanced chemiluminescence (ECL) immunoblotting detection reagent. The band intensity was measured using ImageJ software (National Institute of Health, USA)

## CaM sepharose precipitation

Cells were washed once with ice-cold wash buffer (40 mM HEPES [pH 7.4], 150 mM NaCl), and lysed in ice-cold lysis buffer (40 mM HEPES [pH 7.4], 150 mM NaCl, 0.3% CHAPS or 1% NP-40 or 1% Triton-X100, phosphatase inhibitor cocktail (Sigma) and protease inhibitor cocktail (Roche). The cell debris was removed by centrifugation at 13,000 rpm for 10 min in a microfuge. Five percent of the supernatant was reserved as 'input', and the rest of the supernatant was equally divided into two groups: one contained 1 mM $CaCl_2$, and another one contained 5 mM EGTA. The lysates were incubated with pre-washed CaM sepharose (GE Healthcare) at 4°C for 2 hr with rotation. The beads were washed with $CaCl_2$ (1 mM) or EGTA (5 mM) -containing CHAPS (0.3% ) buffer 3 times, and boiled at 95°C for 5 min. Elution of the protein from CaM sepharose was subjected to immunoblotting to analyze the recovery of indicated proteins or peptides.

## Immunoprecipitations and in vitro kinase assay

Cells were washed once with ice-cold wash buffer (40 mM HEPES [pH 7.4], 150 mM NaCl), and lysed in ice-cold CHAPS buffer (40 mM HEPES [pH 7.4], 150 mM NaCl, 2 mM EDTA, 0.3% CHAPS, phosphatase inhibitor cocktail and protease inhibitor cocktail). The cell debris was removed by centrifugation at 13,000 rpm for 10 min in a microfuge. The soluble fractions of cell lysates were mixed with anti-raptor antibody (4 µg/10 cm dish, Life Technology, Cat. 42–4000, RRID: AB_2533523), and the mixtures were incubated with rotation for 1.5 hr at 4°C. 80 µl of a 50% slurry of protein A/G plus-sepharose (Santa Cruz Biotechnology) was then added and the incubation continued for an additional 1 hr. Immunoprecipitates were washed twice with ice-cold CHAPS buffer, and once with mTOR kinase buffer (25 mM HEPES [pH 7.4], 50 mM KCl, 10 mM $MgCl_2$). The kinase assays were performed as previously described (*Kim et al., 2002*). CaM (2 µM) or/and $CaCl_2$ (1 mM) were added into the kinase reaction as indicated. CMDZ (8 µM) or Torin 1 (100 nM, Cayman Chemical) was

incubated with the reaction mixtures 10 min prior to initiating the reaction by addition of 250 μM ATP (Sigma). The phosphorylation states of S6K or 4EBP1 were detected by immunoblotting.

## Real-time qPCR

HEK293T, HUVEC and A549 cells were transduced with lentivirus carrying scramble shRNA or indicated shRNA. Total RNA was collected using RNeasy Mini Kit (QIAGEN). cDNA was generated with SuperScript III First-Strand kit (Invitrogen), and real-time PCR was carried out using TaqMan Universal Master Mix II (Life Technologies). Real-time PCR primers and probes were from Thermo Fisher Scientific: MCOLN1 FAM (Hs01100653_m1), TPCN2 FAM (Hs01552063_m1). Human GAPDH VIC (Hs02758991_g1) was used as an endogenous control.

## cDNA manipulations and mutagenesis

Myc-mTOR (Addgene plasmid # 1861), pRK5-HA GST RagA 66L (Addgene plasmid # 19300), pRK5-HA GST RagC 75L (Addgene plasmid # 19305) and HA GST PreScission p70 S6K1 (Addgene plasmid # 15511) were gifts from David Sabatini. pcDNA3-FLAG-Rheb-N153T (Addgene plasmid # 19997) was a gift from Fuyuhiko Tamanoi. pcDNA-CaM was a gift from David Yue. TRPML1-HA (Addgene plasmid # 18825) was a gift from Craig Montell. EGFP-Rab7A Q67L (Addgene plasmid # 28049) and EGFP-Rab7A T22N (Addgene plasmid # 28048) were gifts from Qing Zhong.

FLAG-tagged Rheb$^{N153T}$ was amplified by PCR and cloned into the EcoRI site of pLVX-AcGFP-N1 vector. GST-tagged 4EBP1 was amplified by PCR and cloned into a pDEST15-based vector. FLAG-tagged CaM was amplified by PCR and cloned into a pGEX6-based vector. TRPML1 was amplified by PCR and cloned into a pEGFP-based vector. All ligations were performed with Infusion Kit (Clontech Laboratories, Inc.) according to the manufacture's instruction. After sequence verification, these plasmids were used, as described below, in transient cDNA transfections, bacterial protein expression or to produce the lentiviruses needed to generate cell lines stably expressing the proteins.

## cDNA transfection-based experiments

For transfection experiments, HEK293T cells were seeded in 6-well plates or 6 cm culture dishes. After 24 hr, cells were transfected with the pRK5-based cDNA expression plasmids indicated in the figures (500 ng of truncated mTOR fragments; 200 ng HA-GST-tagged RagA 66L or RagC 75L, 1000 ng of EGFP-tagged TRPML1, and same amount of proper empty vectors) using Lipofectamine 2000 (Life Technology) according to the manufacturer's instructions.

## Preparation of p70S6K1, GST-4EBP1 and FLAG-CaM for Use in mTORC1 Kinase Assays

HA-GST-PreScission-p70 S6K1 was transfected into HEK293T cells as described above, and after 48 hr the cells were treated with 20 μM LY294002 for 1 hr prior to cell harvesting and lysis. HA-GST-PreSciss-S6K1 was purified as described (*Burnett et al., 1998*). The purified protein was stored at −20°C in 20% glycerol.

GST-fused 4EBP1 protein was expressed and purified from BL21 (DE3) *Escherichia coli*. Bacteria were grown to an OD of 0.8 and induced for 16 hr at 18°C with 0.5 mM IPTG (American Bioanalytical). Bacteria were pelleted, and lysed in ice-cold PBS containing 1% Triton X-100, 1mg/mL lysozyme (Sigma-Aldrich) and protease inhibitor cocktail by sonication. Cell debris was cleared by centrifugation. The supernatant was mixed with pre-equilibrated glutathione sepharose 4B resin for 1 hr at 4°C with rotation. After gentle centrifugation, GST-4EBP1 was eluted by 10 mM reduced glutathione, and the protein sample was desalted by PD-10 desalting columns and then eluted by the elution buffer (150 mM NaCl, 40 mM HEPES [pH 7.4]). The purified protein was stored at −20°C in 20% glycerol.

GST-FLAG-CaM protein was expressed and purified as GST-4EBP1. GST tag was removed by PreScission (GE Healthcare) according to the manufacturer's instruction. The purified protein was stored at −20°C in 20% glycerol.

## Mammalian lentiviral shRNAs

TRC lentiviral shRNAs targeting hTRPML1, mTOR, hVps34 and raptor were obtained from Sigma. The TRC number for each shRNA is as follows:

Human mTOR shRNA #1: TRCN0000038677
Human mTOR shRNA #2: TRCN0000039785
Human raptor shRNA #1: TRCN0000039772
Human raptor shRNA #2: TRCN0000010415
Human TRPML1 shRNA #1: TRCN0000083297
Human TRPML1 shRNA #2: TRCN0000083296
Human hVps34 shRNA #1: TRCN0000037794
Human hVps34 shRNA #2: TRCN0000037795
Human TPC2 shRNA #1: TRCN0000043919
Human TPC2 shRNA #2: TRCN0000043921
Human P2X4 shRNA #1: TRCN0000044960
Human P2X4 shRNA #2: TRCN0000044962

## Lentivirus production and cell transduction

HEK293T cells were seeded in 15-cm culture dishes. When 50–70% confluent, the cells were co-transfected with 9 µg lentiviral vector (empty, lentiviral vector containing sequences expressing indicated proteins, scramble shRNA or shRNA targeting indicated proteins) + 6 µg pspAX$_2$ + 3 µg pMD$_2$G using lipofectamine 2000 according to manufacturer's instructions. After 24 hr, 48 hr and 72 hr, the supernatants were harvested, respectively, and concentrated using PEG6000 as described before (*Kutner et al., 2009*). The concentrated virus was stored at $-80°$C.

HEK293T or HUVEC cells were seeded in 10 cm culture dishes. When cells were 40% confluent, concentrated lentiviral solutions were added into the cell culture medium. After 48 hr, cells were treated with antibiotics to select transduced cells.

## Immunostaining

Cells were fixed with 4% (wt/vol) paraformaldehyde in PBS for 20 min at room temperature (RT). After wash, cells were permeabilized by PBS/0.5% Triton X-100 and incubated at RT for 10 min. After blocking, cells were incubated with anti-mTOR antibody (1:150, Cell Signaling Technology, Cat. 2983) at 4°C overnight, followed by incubating with Alexa Fluor 568 (1:500, Life technologies, Cat. A11011) for 1 hr at RT. Images were captured using a Zeiss LSM 700 confocal microscope.

## Data analysis

All graphs were created using GraphPad Prism software, and statistical analysis was performed with GraphPad Prism. Data are presented as the mean ± s.d. Two-tail t-test statistical comparisons were made using ANOVA. A *P* value < 0.05 was considered statistically significant. No statistical method was used to predetermine sample size. The experiments were not randomized. The investigators were not blinded to allocation during experiments.

# Acknowledgements

This work was supported in part by the National Cancer Institute (R01CA184103), the Flight Attendant Medical Research Institute (JOL), and the Johns Hopkins Institute for Clinical and Translational Research (ICTR), which is funded in part by Grant Number UL1 TR 001079. We are grateful to Dr. Laixi Wang and Hui Cai for technical advice. We thank Drs. Sarah Head and Zufeng Guo for technical help and support.

# Additional information

### Funding

| Funder | Grant reference number | Author |
| --- | --- | --- |
| National Institutes of Health | R01CA184103 | Ruo-Jing Li<br>Jing Xu<br>Chenglai Fu<br>Jing Zhang<br>Yujun George Zheng<br>Hao Jia |

| | Jun O Liu |
| --- | --- |
| Flight Attendant Medical Research Institute | Ruo-Jing Li<br>Jing Xu<br>Chenglai Fu<br>Jing Zhang<br>Yujun George Zheng<br>Hao Jia<br>Jun O Liu |

The funders had no role in study design, data collection and interpretation, or the decision to submit the work for publication.

## Author contributions

R-JL, Conception and design, Acquisition of data, Analysis and interpretation of data, Drafting or revising the article; JX, CF, JZ, YGZ, HJ, Acquisition of data, Analysis and interpretation of data; JOL, Conception and design, Analysis and interpretation of data, Drafting or revising the article

## Author ORCIDs

Chenglai Fu, http://orcid.org/0000-0003-4300-9948
Jun O Liu, http://orcid.org/0000-0003-3842-9841

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
