## [Decision Letter]

Thank you for submitting your article "Regulation of mTORC1 by Lysosomal Calcium and Calmodulin" for consideration by *eLife*. Your article has been reviewed by two peer reviewers, including Jonathan S Marchant (Reviewer #3), and the evaluation has been overseen by a Reviewing Editor and Kevin Struhl as the Senior Editor.

The reviewers have discussed the reviews with one another and the Reviewing Editor has drafted this decision to help you prepare a revised submission.

All reviewers agree that the discovery of a TRPML-mediated lysosomal Ca^2+^ release mechanism that is necessary for MTORC1 activation is a highly significant advance, and your data are consistent with this conclusion. However, there are a few major concerns that would require confirmation with additional data to substantiate this conclusion and publication in *eLife*:

1) Specificity of the effects you observe for TRPML1: As TRPML1 knockdown is well known to dysregulate the structure of the endolysosmal system, would similar effects occur if other endolysosomal channels were impaired – TPCs (also a MTORC partner), P2X receptors? Are similar effects seen e.g. with state locked rab constructs?

2) Ca^2+^ signaling autonomous to the lysosome: To substantiate the model for a localized acidic Ca^2+^ store, effect compounds that more selectively target acidic Ca^2+^ stores – e.g. bafilomycin, GPN, NAADP-AM – need to be used in experiments. Also, there is little supporting evidence from immunolocalization/imaging to address conditions of mTORC recruitment to TRPML1 positive vesicles versus TRPML1 activity?

3) Signaling effects of VPS34: The effect on S6-P, 4EBP-P, and Akt-473-P after knockdown of VPS34 +/- the CMDZ should be tested. Also, the role of mTOR Complex 2 is not probed directly throughout, because Akt 308-P rather than Akt-473-P is probed (Figure 3). The Akt-473-P blot should be included wherever possible.

---

## [Author Response]

*[…] All reviewers agree that the discovery of a TRPML-mediated lysosomal Ca^2+^ release mechanism that is necessary for MTORC1 activation is a highly significant advance, and your data are consistent with this conclusion. However, there are a few major concerns that would require confirmation with additional data to substantiate this conclusion and publication in eLife:*

*1) Specificity of the effects you observe for TRPML1: As TRPML1 knockdown is well known to dysregulate the structure of the endolysosmal system, would similar effects occur if other endolysosomal channels were impaired – TPCs (also a MTORC partner), P2X receptors? Are similar effects seen e.g. with state locked rab constructs?*

We conducted experiments to address the specificity of the effects of TRPML1. As TPC2 and P2X4 receptors are mainly localized on lysosomes, we knocked down TPC2 and P2X4 using two different shRNA sequences for each. We also used scramble shRNA, shRNA targeting TRPML1 or BAPTA-AM treatment as controls. As shown in revised Figure 1—figure supplement 1 (left panel), the knockdown efficiency of the two shRNA sequences targeting human TPC2 was about 60% – 70% as judged by q-PCR, consistent with the validation data provided by Sigma-Aldrich from which we purchased the shRNA constructs. The knockdown of TPC2 or P2X4 (right panel) did not significantly inhibit the phosphorylation of S6K, while the knockdown of TRPML1 and treatment with BAPTA-AM did, which is consistent with our previous results. In addition, we previously showed that thapsigargin reversed the inhibition of phosphorylation of S6K by TRPML1 knockdown. These pieces of evidence suggest that the effect we observed from TRPML1 knockdown was not due to the dysregulation of the endolysosomal structure. However, we cannot completely rule out the possibility that TPC2 or P2X4, as lysosomal calcium channels, may play a role in regulating mTORC1, given that the knockdown was not complete.

As per reviewers’ suggestion, we also overexpressed constitutively active Rab7A (Q67L) and dominant negative Rab7A (T22N) in HEK293T cells, given that Rab7A plays an important role in positively regulating autophagy (Jäger S et al., Journal of cell science 2004, 117, 4837-4848; Hyttinen JM et al., Biochimica et biophysica acta 2013, 1833, 503-510). As shown in revised Figure 2—figure supplement 2, neither of overexpression of constitutively nor dominant negative Rab7A affected mTORC1 signaling, while overexpression of TRPML1 plus MLSA1 treatment stimulated the phosphorylation of S6K, indicating that the activation of mTORC1 by TRPML1 stimulation was not due to the upregulation of autophagy. We have included these results in our revised manuscript.

*2) Ca^2+^ signaling autonomous to the lysosome: To substantiate the model for a localized acidic Ca^2+^ store, effect compounds that more selectively target acidic Ca^2+^ stores – e.g. bafilomycin, GPN, NAADP-AM – need to be used in experiments. Also, there is little supporting evidence from immunolocalization/imaging to address conditions of mTORC recruitment to TRPML1 positive vesicles versus TRPML1 activity?*

Bafilomycin A1 is an inhibitor of vacuolar-type H^+^-ATPase, and it prevents the re-acidification of lysosomes, therefore depletes lysosomal calcium. We pretreated HEK293T cells with bafilomycin for 1 hour, and continuously treated the cells with or without MLSA1 for another hour. As shown in revised Figure 2, bafilomycin pretreatment largely abolished the stimulation of mTOR by MLSA1, indicating that the stimulative effect by MLSA1 depends on lysosomal calcium storage. Glycyl-L-phenylalanine 2-naphthylamide (GPN) is a Cathepsin C substrate that induces osmotic lysis of lysosomes and depletes lysosomal calcium. We saw similar effect after GPN pretreatment. These results are consistent with previous report that treatment of cells with bafilomycin A1 or GPN abolished the response of TRPML1 to MLSA1 stimulation (Shen D et al., Nat Commun. 2012, 3:731).

NAADP-AM is reported to bind to TPC2 and induced Ca^2+^ release through TPC2 in TPC2 overexpressed HEK293 cells. However, wild-type cells showed only small and short-lived lysosomal calcium release from native TPC2, and lacked both the ramp-like phase and the secondary transient (Calcraft PJ et al., Nature 2009, 459, 596-600). We did not observe stimulative effect by NAADP treatment in wild type HEK293T cells, likely due to the smaller and shorter calcium release.

Author response image 1.HEK293T cells were treated with NAADP (5 μM) for 1 hr, and were lysed and subjected to immunoblotting.**DOI:**
http://dx.doi.org/10.7554/eLife.19360.015

To test if mTOR is recruited onto TRPML1 positive vesicles, we performed immunofluorescence assay. Because there is no reliable anti-TRPML1 antibody for immunostaining available, we visualized TRPML1 by transfecting HEK293T cells with EGFP-TRPML1 expression plasmid. As shown in Figure 2—figure supplement 1, when cells were deprived with amino acids, mTOR failed to cluster and was not colocalized with EGFP-TRPML1. However, when the cells were restimulated with amino acids, mTOR clustered and EGFP-TRPML1 largely colocalized with mTOR, indicating that mTOR was recruited to lysosomes and colocalized with TRPML1. These observations are consistent with previous report that TRPML1 colocalized with lysosomes (Cheng X, et al., FEBS Lett. 2010, 584: 2013–2021; Li X et al., Nat Cell Biol. 2016, 18:404-17), and amino acids stimulate the translocation of mTORC1 onto lysosomes with the help of Rag GTPase (Zoncu R et al., Science 2011, 334: 678–683; Sancak Y et al., Cell 2010,141:290-303). This result supports our model in that mTOR is recruited to the lysosome and colocalize with TRPML1 that releases lysosomal calcium locally to activate lysosomally localized mTOR upon stimulation with animo acids.

*3) Signaling effects of VPS34: The effect on S6-P, 4EBP-P, and Akt-473-P after knockdown of VPS34 +/- the CMDZ should be tested. Also, the role of mTOR Complex 2 is not probed directly throughout, because Akt 308-P rather than Akt-473-P is probed (Figure 3). The Akt-473-P blot should be included wherever possible.*

To address the first part of this question, we knocked down hVps34 and treated the wild type or knockdown cells with CMDZ or vehicle control. As shown in revised Figure 4—figure supplement 1, the depletion of hVps34 did not affect the sensitivity of mTOR to CMDZ treatment (Figure 4—figure supplement 1), indicating that the regulation of mTOR by hVps34 is not dependent on CaM. Another study by an independent group also suggested that the activity of hVps34 was independent of Ca^2+^/CaM (Yan Y et al., Biochem J 2009, 417:747-55). However, consistent with previous reports (Byfield MP et al., J Biol Chem. 2005, 280:33076-82; Yoon MS et al., Cell Rep. 2016, 16:1510-7; Nobukuni T et al., Proc Natl Acad Sci U S A. 2005, 102:14238-43.), knockdown of hVps34 largely abolished mTORC1 activation by amino acids stimulation (Figure 4—figure supplement 1).

For the second part of the question, we agree with the reviewers on the need to assess p-Akt (473) in Figure 3. We have added the blot of p-Akt (473) to our revised Figure 3, and we also included p-Akt (473) as much as possible in other experiments (Figure 1—figure supplement 1, Figure 2—figure supplement 2, Figure 4—figure supplement 1).